# Magnetization reversal through an antiferromagnetic state

**Somnath Ghara** [1] ✉, **Evgenii Barts** [2], **Kirill Vasin** [1,3], **Dmytro Kamenskyi**[1], **Lilian Prodan**[1], **Vladimir Tsurkan**[1,4], **István Kézsmárki** [1], **Maxim Mostovoy** [2] & **Joachim Deisenhofer** [1]

Magnetization reversal in ferro- and ferrimagnets is a well-known archetype of non-equilibrium processes, where the volume fractions of the oppositely magnetized domains vary and perfectly compensate each other at the coercive magnetic field. Here, we report on a fundamentally new pathway for magnetization reversal that is mediated by an antiferromagnetic state. Consequently, an atomic-scale compensation of the magnetization is realized at the coercive field, instead of the mesoscopic or macroscopic domain cancellation in canonical reversal processes. We demonstrate this unusual magnetization reversal on the Zn-doped polar magnet $Fe_2Mo_3O_8$. Hidden behind the conventional ferrimagnetic hysteresis loop, the surprising emergence of the antiferromagnetic phase at the coercive fields is disclosed by a sharp peak in the field-dependence of the electric polarization. In addition, at the magnetization reversal our THz spectroscopy studies reveal the reappearance of the magnon mode that is only present in the pristine antiferromagnetic state. According to our microscopic calculations, this unusual process is governed by the dominant intralayer coupling, strong easy-axis anisotropy and spin fluctuations, which result in a complex interplay between the ferrimagnetic and antiferromagnetic phases. Such antiferro-state-mediated reversal processes offer novel concepts for magnetization control, and may also emerge for other ferroic orders.

Magnetization reversal in ordinary ferro- and ferrimagnets occurs via the expansion of the energetically favored domain state at the expense of the unfavoured domain state, leading to a cancellation of the total magnetization at the coercive field. Magnetization reversal processes are essential in many applications. The on-demand control of domains upon magnetization reversal is a hotbed of new developments for spintronics[1–4]. The mutual coupling of magnetization and polarization in magnetoelectric and multiferroic materials can be exploited to reverse magnetization by electric field and for electric read-out of the magnetic state[5,6].

Recently, the honeycomb antiferromagnets $A_2Mo_3O_8$ (A=Mn,Fe,Co,Ni,Zn) have emerged as a versatile material class with a hexagonal structure in the polar space group $P6_3mc$[7–17]. These compounds exhibit optical magnetoelectric effects such as directional dichroism[18,19] and giant thermal Hall effects[20,21], and the control of magnetization and polarization can be achieved in magnetic fields of a few Tesla[22,23] or even by ultrafast modulation via laser pulses[24]. Particular attention has been drawn to $Fe_2Mo_3O_8$ with a Néel temperature $T_N$=60 K, below which a collinear antiferromagnetic (AFM) spin order of the magnetic $Fe^{2+}$

[1]Experimentalphysik V, Center for Electronic Correlations and Magnetism, Institute for Physics, University of Augsburg, D-86135 Augsburg, Germany. [2]Zernike Institute for Advanced Materials, University of Groningen, Nijenborgh 4, 9747 AG Groningen, The Netherlands. [3]Institute for Physics, Kazan (Volga region) Federal University, 420008 Kazan, Russia. [4]Institute of Applied Physics, Moldova State University, MD-2028 Chişinău, Republic of Moldova. ✉e-mail: somnath.ghara@physik.uni-augsburg.de

ions coaligns with the polarization along the crystallographic *c*-axis as shown in Fig. 1a. The $Fe^{2+}$ ions occupy tetrahedrally coordinated sites (A-site) and octahedrally coordinated ones (B-site). Within each of the hexagonal layers (Fig. 1c) the inequality of the orbital contributions to the magnetic moments of A- and B-site ions with spin $S = 2$ results in finite magnetizations $\mathbf{M}_{1,2}$ of each layer with opposite sign for adjacent layers (green arrows in Fig. 1a). Such an AFM spin configuration can be characterized by the AFM Néel vectors for each sublattice of $Fe^{2+}$, e.g., $\mathbf{L}_A = \frac{1}{2}(\mathbf{M}_{1A} - \mathbf{M}_{2A})$ with sublattice magnetizations $\mathbf{M}_{1A}$ and $\mathbf{M}_{2A}$ for the two different A-sites. The AFM order parameter is then given by $\mathbf{L} = \mathbf{L}_A + \mathbf{L}_B = \frac{1}{2}(\mathbf{M}_1 - \mathbf{M}_2)$. Upon application of a magnetic field along the *c*-axis, a transition to a ferrimagnetic (FiM) phase occurs, where the two anti-aligned sublattices are formed by A-site and B-site ions, respectively, corresponding to a flip of the layer magnetization in every second layer, as shown in Fig. 1b[22,23]. This complies with a clearly dominating AFM intralayer exchange coupling $J_∥$ (Fig. 1c) in comparison with the weaker interlayer couplings $J_{AA}$ and $J_{BB}$ (Fig. 1d). The order parameter of the FiM state can be expressed as the sum of the two sublattice contributions $\mathbf{M}_{A,B}$ or of the layer magnetizations $\mathbf{M}_{1,2}$ as $\mathbf{M} = \mathbf{M}_A + \mathbf{M}_B = \mathbf{M}_1 + \mathbf{M}_2$.

The FiM state was shown to exhibit a linear magnetoelectric effect[23] and can be stabilized by substituting Fe by non-magnetic Zn, which preferably occupies the tetrahedral A-sites[8,9,23,25,26]. The persistence of the field-induced FiM state in Zn-doped $Fe_2Mo_3O_8$ upon decreasing and reversing the field was reported beforehand[23,27], but the origin of this metastable state has not been addressed previously. The Néel temperature in the system $Fe_{1.86}Zn_{0.14}Mo_3O_8$ is reduced to 53 K as indicated by the peak in the magnetic susceptibility shown in Fig. 1e, while the polarization shows an increase at the magnetic ordering and saturation-like behavior in the AFM phase comparable to pure $Fe_2Mo_3O_8$[23]. The FiM state at this Zn concentration persists as a metastable state in zero magnetic field below about 40 K, i.e., the remanent magnetization during hysteretic cycling remains finite, as seen in Fig. 2a, b[27].

Here, we report an unconventional magnetization reversal observed in the field-induced FiM state of lightly Zn-doped $Fe_2Mo_3O_8$. We find that reversing the magnetization requires the resurrection of the AFM state, which is evidenced by a clear peak in the static polarization and the reemergence of a THz excitation as a characteristic footprint of the AFM state. Our microscopic theoretical approach can explain the origin of the FiM metastable state and the resurrection of the AFM phase upon magnetization reversal. The present study offers a new pathway for magnetization reversal, where instead of the conventional mesoscopic and macroscopic domain cancellation, an atomic-scale compensation of the magnetization via the emergence of an AFM state is observed at the coercive field.

## Results and discussion
### An unusual magnetization reversal

In this section, we discuss how the reoccurrence of the AFM state at the magnetization reversal is reflected in the polarization. In Fig. 2a, b we show the magnetization $M(H)$ (dashed lines) and polarization $P(H)$ (solid lines) measured as a function of the external magnetic field along the *c*-axis at $T = 20$ K and $T = 13$ K, respectively. The magnetization curves starting at zero magnetic field (black dashed lines) in the pristine AFM state show a transition to the FiM state in the shaded field range, and upon reversing the magnetic field, a typical FiM-like hysteresis shows up with coercive fields of the order of the critical fields of the AFM-to-FiM transition. This is in agreement with previously reported data[27] and samples with similar Zn concentrations[23]. The polarization shows a strong decrease upon the transition from the AFM to the FiM state, and a linear magnetoelectric effect occurs in the FiM state (shown clearly in Supplementary Fig. 1), both observations in line with literature[23]. Intriguingly, whenever the magnetization starts to deviate from its saturation values $\pm M_S$ during the cycling, a strong increase in polarization occurs, reaching its maximum at the coercive field (see e.g., shaded area in Fig. 2a). Such an unusual behavior has not been observed beforehand in Zn-doped $Fe_2Mo_3O_8$ or other FiM multiferroics exhibiting a linear magnetoelectric effect[28].

It is important to note that both the critical and coercive fields and the peak height of the polarization depend on temperature, as can be seen by the comparison of Fig. 2a, b. At 20 K the maximal value of the polarization at the coercive field is about half of the one in the pristine AFM phase (Fig. 2a), but at 13 K the value is already considerably reduced. At 30 K the maximal polarization value at the coercive field almost reaches the value of the pristine AFM phase (see Supplementary Fig. 2). We regard this as evidence that in the vicinity of the coercive fields, when the mono-domain FiM phase usually turns to an equal share of up and down domains with volume fractions $x_{fim\uparrow}(H,T)$ and $x_{fim\downarrow}(H,T)$, respectively, a significant fraction of the sample volume, denoted as $x_{afm}(H, T)$, exhibits the properties of the pristine AFM state with high polarization values. The AFM fraction assisting the magnetization reversal emerges now as a metastable state.

Consequently, we analyze the magnetic-field dependent magnetization and polarization data assuming that the entire sample volume is distributed between three fractions only, i.e., $x_{afm} + x_{fim\uparrow} + x_{fim\downarrow} = 1$. The magnetization is then solely determined by the FiM volume fractions as $M(H) = M_S(x_{fim\uparrow} - x_{fim\downarrow})$, while the polarization bears contributions of all three magnetic volume fractions,

$$P(H) = x_{afm}P^0_{afm} + P^0_{fim}(x_{fim\uparrow} + x_{fim\downarrow}) + \alpha H(x_{fim\uparrow} - x_{fim\downarrow}). \tag{1}$$

Since the compound is pyroelectric, the electric polarization of the magnetic field-induced FiM state is different from that in the AFM

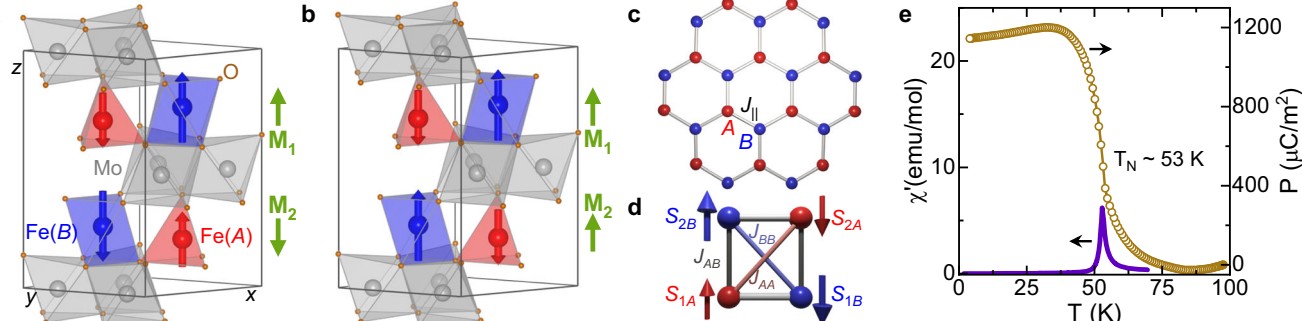

**Fig. 1 | Crystal structure and magnetic order in $Fe_{2-x}Zn_xMo_3O_8$. a**, **b** Unit cell of $Fe_2Mo_3O_8$ in the AFM state and the FiM state, with tetrahedral (*A*) and octahedral (*B*) sites shown in red and blue, respectively. The greeen arrows indicate the overall magnetization of each layer. **c** Hexagonal layer formed by the Fe sites with dominant AFM exchange coupling $J_∥$. **d** Exchange couplings $J_{AA}, J_{BB}$ and $J_{AB}$ between the Fe ions at *A* and *B* sites of adjacent layers in the AFM state. **e** Temperature dependence of the real part of the magnetic *ac* susceptibility $\chi'$ (solid line) and polarization *P* (open symbols) in $Fe_{1.86}Zn_{0.14}Mo_3O_8$.

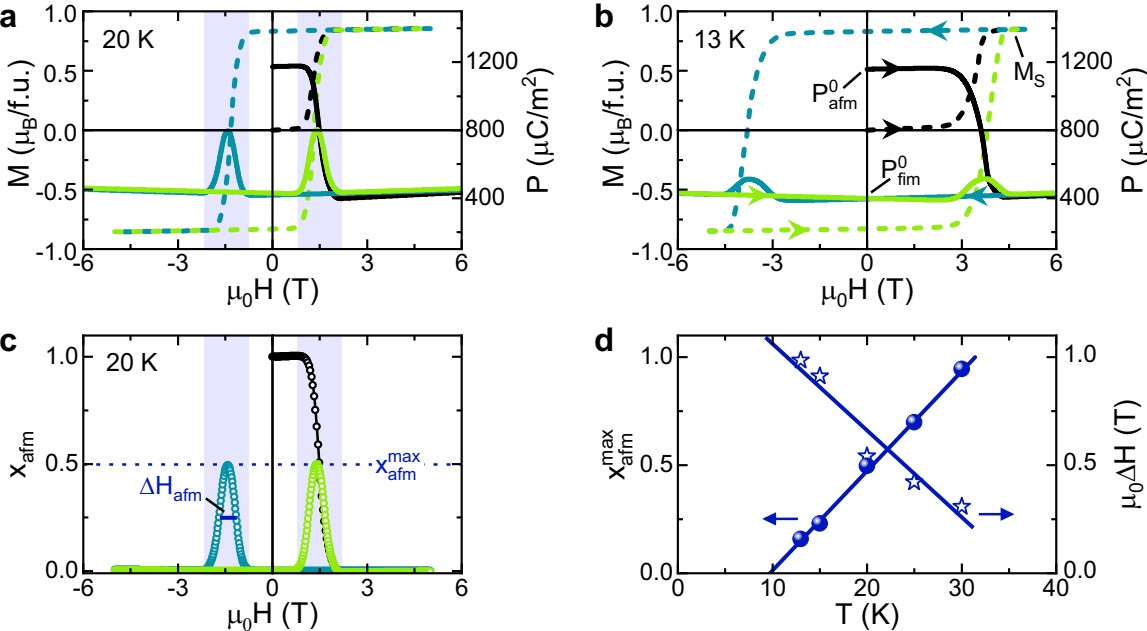

**Fig. 2 | Signatures of the resurrection of the AFM state in magnetization and polarization. a, b** Magnetic field-dependent magnetization $M$ (dashed lines) and polarization $P$ (solid lines) along the $c$ axis at 20 K and 13 K, respectively. **c** Magnetic field-dependent antiferromagnetic volume fraction ($x_{afm}$) at 20 K extracted from magnetization and polarization data using Eq. (2). The vertical shading in (**a** and **c**) indicates the field range, where the pristine AFM state reappears upon magnetization reversal. **d** Temperature dependence of the maximum values of $x_{afm}$ and of the full-width-half maximum values $\Delta H_{afm}$. The solid lines are to guide the eyes.

ground state and may vary with temperature for both states. To include this effect, we experimentally determine the polarization of the pristine antiferromagnet in zero magnetic field $P_{afm}^0$ and the corresponding polarization $P_{fim}^0$ of the FiM phase upon lowering the field to zero after reaching the mono-domain FiM states with $M(H) = \pm M_S$ at each temperature (see Fig. 2b). Moreover, time-reversal invariance implies that the polarization is independent of the sign of **M** or the AFM Néel vector **L**, which results in the first two terms in Eq. (1). The third term represents the contribution due the strong linear magnetoelectric effect of the FiM state[23,27], where the sign of the linear magnetoelectric coefficient $\alpha$ is different for the two FiM volume fractions. Upon substituting $(x_{fim\uparrow} - x_{fim\downarrow}) = M(H)/M_S$, we can extract the AFM volume fraction

$$x_{afm}(H) = \frac{1}{P_{afm}^0 - P_{fim}^0}\left[P(H) - P_{fim}^0 - \alpha H \frac{M(H)}{M_S}\right], \quad (2)$$

from our experimental data of $P(H)$ and $M(H)$. The only parameter which had to be determined by fitting the $P(H)$ curves in the FiM regime is $\alpha$, which was derived in the corresponding linear regimes (a possible contribution $\propto H^2$ was found to be negligible, see Supplementary note 1).

The result for $T = 20$ K is shown in Fig. 2c. The values of $x_{afm}(H)$ yield symmetric peaks centered at the coercive fields. The corresponding maximum values $x_{afm}^{max}$ and the full-width-half-maximum width $\Delta H_{afm}$ of the curves are shown in Fig. 2d for all investigated temperatures (see Supplementary Fig. 3 for details). While $x_{afm}^{max}$ decreases linearly with decreasing temperature and extrapolates to zero at around 10 K, the linear increase of $\Delta H_{afm}$ may reflect different kinetics of the metastable states in the vicinity of the magnetization reversal. This is in agreement with the absence of any macroscopic polarization peak at the magnetization reversal below 10 K.

To corroborate the above scenario derived from *dc* magnetization and polarization measurements, we will discuss in the following the dynamic fingerprints of the magnetic volume fractions in the THz frequency regime.

## Probing magnetic phases via THz spectroscopy

The resurrection of the pristine AFM phase and its coexistence with the FiM fraction upon magnetization reversal can also be revealed in the THz spectra by investigating the field evolution of the characteristic elementary excitations of the two magnetic phases, which were previously identified[27]. In Fig. 3b we show the THz absorption spectra (red colors) at 25 K for light polarization $E^\omega \| a$ for several magnetic fields during the virgin magnetization curve with the external magnetic field $H \| c$ as indicated by the same colors in the $M$-$H$-diagram in Fig. 3a. The spectrum of the AFM ground state at zero field and the one of the saturated FiM state in a field of 2 T serve as benchmarks for the virgin AFM and mono-domain FiM states with $x_{afm} = 1$ and $x_{fim\uparrow} = 1$, respectively. The absorption spectra shown in Fig. 3c correspond to reversed magnetic fields of the hysteretic cycle (green curves) and show the evolution of the magnetic phases upon approaching the magnetization reversal at about −1 T and reaching again a fully saturated FiM state at − 2 T with $x_{fim\downarrow} = 1$. The spectrum of the pristine AFM state is characterized by one distinct spectral feature in this configuration - the narrow electric-dipole active absorption mode at 44 cm⁻¹ (AFM mode) shown in Fig. 3b, which was identified previously as a clear fingerprint of the AFM state both in pure $Fe_2Mo_3O_8$[29] and in $Fe_{1.86}Zn_{0.14}Mo_3O_8$[27]. The mode at 44 cm⁻¹ is clearly absent in the spectra of the saturated FiM states for $H = \pm 2$ T, where only one excitation at about 83 cm⁻¹ (FiM mode) is observed as a characteristic fingerprint of the FiM state[27].

Upon lowering and reversing the external magnetic field from the saturated FiM state at 2 T no significant changes in the absorption spectra occur as long as the condition $M(H) = M_S$ holds (spectra not shown). Intriguingly, on approaching magnetization reversal an excitation emerges at the eigenfrequency of the characteristic AFM mode at 44 cm⁻¹, reaches a maximum in intensity at around $H_c = -1$ T, and decreases in intensity again for $|H| > H_c$ as shown in Fig. 3c. Finally, the mode disappears again when reaching magnetization saturation with $M(H) = -M_S$ on approaching a field of −2 T. Based on the comparison with the properties of the spectrum in the pristine AFM state, we identify this emerging mode with

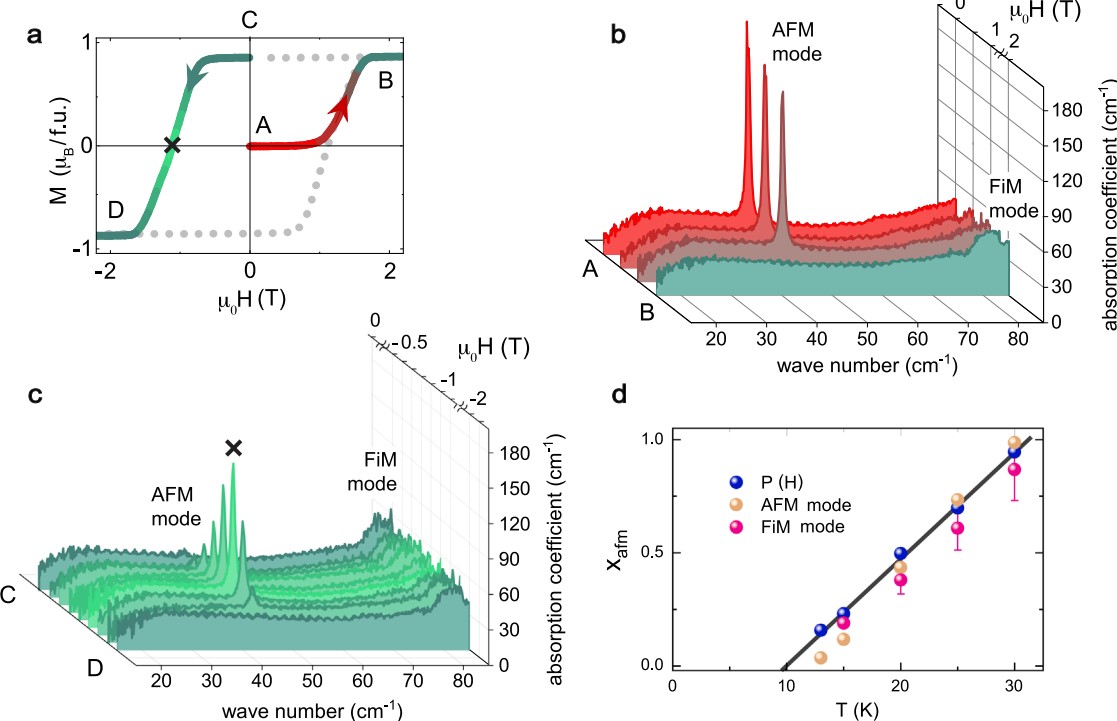

**Fig. 3 | Demonstration of the reappearance of the AFM state upon magnetization reversal via THz spectroscopy. a** Magnetic field-dependent magnetization $M(H)$ at 25 K. Selected segments (A-B and C-D) of the magnetization curve are highlighted by colors to indicate the field regions of the THz data shown in (**b**) and (**c**). **b** THz absorption spectra at 25 K recorded at different magnetic fields ($H\|c$) ranging between the segment A-B of (**a**). **c** THz absorption spectra recorded at different magnetic fields ranging between the segment C-D of (**a**). The THz

spectrum indicated by the symbol × is recorded at the coercive field (see **a**). **d** Temperature dependence of the maximum values of $x_{\mathrm{afm}}$ at the magnetization reversal obtained directly from the integrated intensity of the AFM mode and the FiM mode (via $1 − x_{\mathrm{fim}}$). The error for the values obtained from the broad FiM mode were estimated to be about 15%, while the error for the narrow AFM mode is smaller than 1% and within the symbol size. For comparison, $x_{\mathrm{afm}}$ values obtained from the polarization analysis are reproduced in (**d**). The solid line is a guide to the eye.

the AFM mode: eigenfrequency and linewidth of the two modes coincide and they obey the same electric-dipole selection rule $E^\omega\|a$ (see Supplementary Fig 4). In addition, the intensity of the FiM mode clearly decreases with increasing intensity of AFM mode during magnetization reversal, showing that the THz spectra provide direct information on the coexistence of the AFM and FiM states in this magnetic field range.

We used the integrated intensities of AFM and FiM modes for all spectra to extract the AFM volume fraction $x_{\mathrm{afm}}$ and the FiM one $x_{\mathrm{fim}}$ by normalizing the intensity values to the intensity of the modes in the pristine AFM and the saturated FiM states, respectively (see Supplementary note 2 for details). Note that our THz probe does not distinguish up and down FiM states, therefore $x_{\mathrm{fim}} = x_{\mathrm{fim}\uparrow} + x_{\mathrm{fim}\downarrow}$. The values of $x_{\mathrm{afm}}$ obtained directly from the AFM mode and using $x_{\mathrm{afm}} = 1 − x_{\mathrm{fim}}$ from the FiM mode are shown in Fig. 3d for all measured temperatures, together with the values of $x_{\mathrm{afm}}$ from the above analysis of the magnetization and polarization using Eq. (2). The agreement of the $x_{\mathrm{afm}}$ values from the dynamic THz probe and the static polarization and magnetization values is very good and confirms the unusual resurrection of the AFM ground state as a metastable state during the magnetization reversal of the FiM state.

Having established the magnetization reversal through the AFM state using our experimental observations, we will now discuss a theoretical approach that explains this scenario from a microscopic point of view.

## Microscopic theory of the magnetization reversal

In order to understand the puzzling appearance of the AFM phase with its high polarization at magnetization reversals, we consider the

following spin model

$$
\begin{aligned}
E = J_\| \sum_{\langle i,j\rangle} s_i \sigma_j + J_\perp \sum_{i\in A} s_i \left(\sigma_{i+c/2} + \sigma_{i-c/2}\right) \\
- H\left(\sum_{i\in A} m_A s_i + \sum_{j\in B} m_B \sigma_j\right),
\end{aligned}
\tag{3}
$$

where $s_i = S_i^z / S$ with $S_i^z = 0, \pm1, \pm2$ denoting the $c$-axis projection of the spin of a tetrahedrally coordinated $Fe^{2+}$ ion ($S = 2$) on sublattice A, and $\sigma_j = \pm1$ is an Ising variable describing the strongly anisotropic spins on sublattice B of octahedrally coordinated Fe ions. The first term in Eq. (3) describes the dominating AFM exchange interaction $J_\| > 0$ between neighboring spins in the $ab$-layers (see Fig. 1c) and the second term describes the weak effective ferromagnetic interaction $J_\perp < 0$ between spins in neighboring layers along the vertical AB bonds, resulting from the interplay between the three AFM interlayer interactions $J_{AA}, J_{BB}$ and $J_{AB}$[30] depicted in Fig. 1d. Here, $\sigma_{i\pm c/2}$ denotes the B-spins located above and below the spin on the A-site $i$. The last term is the Zeeman energy for a magnetic field applied along the $c$ axis with the magnetic moments of A and B site spins set to $m_B = 4.5\,\mu_B$ and $m_A = 4.2\,\mu_B$. These values are in agreement with neutron diffraction measurements[9] and reproduce the experimentally observed saturation magnetization $M_s = 0.86\,\mu_B$/f.u. for $x = 0.14$ at low temperature assuming that Zn substitutes Fe on tetrahedral sites[8,9,23,26]. The large deviation of $m_B$ from the spin-only value $4\,\mu_B$ results from the unquenched orbital moment of $Fe^{2+}$ ions on octahedral sites, which also leads to strong single-ion anisotropy along the $c$ axis and allows us to describe B-spins by Ising variables. The relatively weak anisotropy of the A-site ions is neglected.

With this approach we first simulate the temperature and magnetic field dependence of the magnetization and polarization in

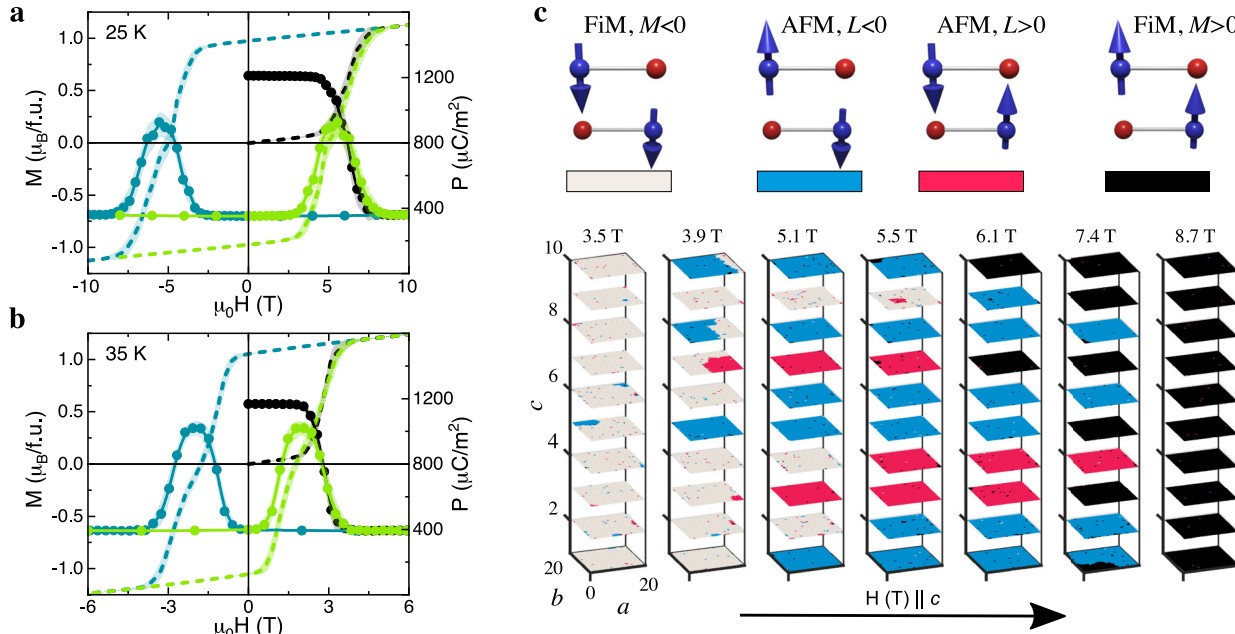

**Fig. 4 | Simulation of spin-state evolution upon magnetization reversal.**
**a**, **b** Magnetic field dependence of calculated magnetization $M$ (dashed lines) and polarization $P$ (solid circles) at 25 K and 35 K, respectively. **c** Snapshots of spin configurations at different magnetic fields at 25 K during a magnetization reversal process from a FiM down-domain state to a FiM up-domain state. Each plaquette corresponds to a unit cell of $Fe_{1.86}Zn_{0.14}Mo_3O_8$, with $M = -1$, $L = 0$ (white), $M = 0$, $L = -1$ (blue), $M = 0$, $L = +1$ (red) and $M = +1$, $L = 0$ (black), where the AFM order parameter $L = (S_1 - S_2)/2$ and the FiM order parameter $M = (S_1 + S_2)/2$ are determined by the $B$-site spins $S_{1,2}$ in the unit cell. Color coding of magnetic orders is shown at the top of the panel. The planes are $ab$ layers stacked along the $c$ axis.

pure $Fe_2Mo_3O_8$ showing the field-induced transition from the AFM to the FiM state (see Supplementary notes 3 and 4 for details of calculations). Supplementary Fig. 5a shows $M(H)$ curves calculated for $J_\parallel = 47$ K, $J_\perp = -1.2$ K, which are in quantitative agreement with experiment[22,23]. The magnetically-induced electric polarization (shown in Supplementary Fig. 5b) is calculated using a microscopic magnetoelectric coupling. The field-induced transition corresponds to a spin-flip transition instead of the spin-flop transition expected for isotropic spins. The net magnetic moment of an $ab$ layer that governs this transition depends on temperature, because the isotropic A-site spins show larger thermal fluctuations than the Ising spins on B-sites. As a result, the net magnetic moment of an $ab$ layer near $T_N$ is larger than at low temperatures[23]. Conversely, the critical field necessary to flip spins just below $T_N$ is significantly smaller than that at low temperatures, which explains the dramatic increase of the spin-flip field upon decreasing temperature in pure $Fe_2Mo_3O_8$[22,23]. The expansion of the boundary of the AFM phase toward higher magnetic fields at low temperatures plays an important role in the emergence of this phase at the magnetization reversals.

In Fig. 4 we show the main results of our calculations with regard to the experimentally observed polarization anomaly upon magnetization reversal in $Fe_{1.86}Zn_{0.14}Mo_3O_8$. Our theory reproduces the hysteretic magnetization and the polarization curves with a peak at the coercive field (Fig. 4a, b). As temperature decreases, the coercive field increases due to the reduction of thermal fluctuations. The substitution of non-magnetic Zn for Fe on tetrahedral sites increases the net magnetic moment, which further stabilizes the FiM state reducing the critical field to 0 just below $T_N$ at $x = 0.14$[27]. To reproduce the strong reduction of the critical field upon doping, we use a much smaller value of the interlayer exchange constant, $J_\perp = -0.2$ K. In fact, the effective ferromagnetic interaction $J_\perp$ is expected to be $x$-dependent, as it results from the interplay between the three AFM interlayer interactions $J_{AA}$, $J_{BB}$ and $J_{AB}$ (as depicted in Fig. 1d)[30]. This interplay sensitively

depends on the removal of magnetic ions from A-sites. The much weaker interlayer exchange coupling $J_\perp$ may be the reason why this phenomenon is so prominent in the present compound $Fe_{1.86}Zn_{0.14}Mo_3O_8$. Since no such signatures of the resurrection of the AFM state have been reported for other compositions in the $Fe_{2-x}Zn_xMo_3O_8$ series[23], the necessary $J_\perp$ for this unusual magnetization reversal may be realized only in a limited range of Zn concentrations.

Figure 4c shows snapshots of the spin configurations during the simulated magnetization reversal that starts from a prepared FiM state with negative saturation magnetization, which remains negative up to +3.5 T. Each plane corresponds to two neighboring magnetic layers of $Fe_{1.86}Zn_{0.14}Mo_3O_8$ and the color denotes the local magnetic ordering in the B-site sublattice described by the calculated local order parameters $L = (S_{B1} - S_{B2})/2 = +1$, $M = (S_{B1} + S_{B2})/2 = 0$ (red), $L = -1$, $M = 0$ (blue), $L = 0$, $M = +1$ (black), and $L = 0$, $M = -1$ (white). The strong intralayer AFM coupling aligns the spins of the A and B sublattices opposite to each other. Therefore, we consider only the B-site sublattice order parameters here. Starting out from a FiM state with negative saturated magnetization (white planes) on the left, planes with non-zero AFM order parameter emerge with increasing field in the vicinity of the coercive field (blue and red planes) before the FiM state with positive magnetization (black planes) is reached at the highest fields.

The nearly uniform color of each plane reflects strong spin correlations in the $ab$ layers (see Fig. 4c). These spin correlations suppress the growth of droplets with the opposite magnetization induced by the magnetic field reversal. The correlation length along the c-axis in the AFM phase is small due to weak interlayer coupling. The slow kinetics of the magnetization switching and the enhanced stability of the AFM phase at low temperatures delay the emergence of the AFM state up to the coercive field. The simulation snapshot in Fig. 4c shows almost complete disappearance of the FiM state in favor of the AFM state, indicating the importance of the

small interlayer coupling. Averaged over many simulations, the maximal fraction of the AFM phase decreases with decreasing temperature (see Fig. 4a, b) in agreement with our polarization and THz data.

The above calculations, supporting nicely the experimental findings, provide the microscopic understanding of how the strong easy-axis anisotropy together with the strong intralayer and weak interlayer couplings causes the unusual reappearance of the AFM state at the magnetization reversal in $Fe_{1.86}Zn_{0.14}Mo_3O_8$.

To summarize, by independent measurements of polarization, magnetization and THz absorption, we discovered a highly unusual magnetization reversal process of the FiM state in Zn-doped $Fe_2Mo_3O_8$, which involves the reappearance of the AFM ground state as a metastable state during the reversal process. Our theoretical simulations nicely reproduce this finding, showing that an Ising-like anisotropy of the octahedrally coordinated Fe spins, small magnetic moment of FiM $ab$ layers and weak interlayer interactions are the main ingredients for the emergence of these metastable spin configurations: The persistence of the FiM state in zero field and the reappearance of the AFM state at the coercive field.

Although Zn-doped $Fe_2Mo_3O_8$ has a unique combination of properties that slow down kinetics of transitions between the FiM and AFM phases and allow for electric detection of these states, the conditions for layer-wise magnetization switching via metastable spin states of distinct characters may be realized in other (quasi) 2D magnetic systems[31–34]. In addition, the fact that ferrimagnetism is associated with the different crystallographic environments of A- and B-site $Fe^{2+}$ ions clearly distinguishes our case from traditional 3D ferrimagnets, such as, e.g., the spinel $FeCr_2S_4$ with two different types of magnetic ions[35–37].

At this point, we want to stress the difference between the honeycomb antiferromagnets $A_2Mo_3O_8$ and the van der Waals magnets. In the latter systems, the coexistence of AFM and ferromagnetic states usually requires a complex material design using nanofabrication, such as mechanical twisting in $CrI_3$[3], whereas the honeycomb antiferromagnets $A_2Mo_3O_8$ (A=Fe,Co,Mn,Ni,Zn) provide a versatile intrinsic toolbox for tuning and controlling different spin configurations through layer-by-layer switching. In fact, a special coexistence of collinear antiferromagnetism with canted ferrimagnetism on adjacent honeycomb layers was recently observed in high-magnetic fields[38]. We believe that similar exotic switching processes, involving metastable states of distinct magnetic order, can be achieved in these compounds by other external stimuli like pressure, voltage or intense light-fields, which will allow to control the growth and decay rates of the spin states and open new pathways for spintronic applications.

## Methods

### Synthesis
Polycrystalline $Fe_{2−x}Zn_xMo_3O_8$ samples were prepared by repeated synthesis at 1000 °C of binary oxides FeO (99.999%), $MoO_2$ (99%), and ZnO in evacuated quartz ampoules aiming for a concentration with $x = 0.2$. Single crystals were grown by the chemical transport reaction method at temperatures between 950 and 900 °C. $TeCl_4$ was used as the source of the transport agent. Large single crystals up to 5 mm were obtained after 4 weeks of transport. The X-ray diffraction of the crushed single crystals revealed a single-phase composition with a hexagonal symmetry using space group $P6_3mc$ and a Zn content corresponding to $x \approx 0.14$. The obtained lattice constants are $a = b = 5.773(2)$ Å and $c = 10.017(2)$ Å[27].

### THz spectroscopy
Temperature and magnetic field dependent time-domain THz spectroscopy measurements were performed on a plane-parallel $ac$-cut single crystal of $Fe_{1.86}Zn_{0.14}Mo_3O_8$. A Toptica TeraFlash time-domain THz spectrometer was used in combination with a superconducting magnet, which allows for measurements at temperatures down to 2 K and in magnetic fields up to ± 7 T. Measurements were performed in Voigt transmission configuration with the magnetic field parallel to the $c$-axis.

### Magnetization measurements
dc and ac magnetization measurements were carried out with a Magnetic Property Measurement System (5 T MPMS-SQUID, Quantum Design). The magnetic field was applied along the $c$ direction of a hexagonal-shaped single crystal with clear top-bottom $ab$ planes.

### Polarization measurements
Electric polarization was obtained by measuring pyroelectric/magnetoelectric current with a Keysight electrometer (model number B2987A). For accessing low temperature and high-magnetic fields, a Physical Property Measurement System (9 T PPMS, Quantum Design) and an Oxford helium-flow cryostat (+14 T) were used. The measurements were carried with a hexagonal-shaped single crystal, where electrical contacts were made on top-bottom $ab$ planes by silver paint and the electric polarization along the $c$ axis was probed. The crystal was mounted in PPMS/cryostat in such a way that the magnetic field could be applied along the $c$ axis and the field was varied with a rate of 100 Oe/sec. Temperature-/magnetic field-dependent electric polarization was obtained by integrating the pyroelectric/magnetoelectric current over measurement time.

### Numerical simulations
The sum over spin projections on tetrahedral sites, $s_i$, in Eq.(3) can be performed analytically, which leaves an effective model of Ising variables $\{\sigma_i\}$ (for more details see Supplementary materials). This effective model was used to calculate the magnetization and antiferromagnetic order parameter, as well as for simulations of hysteresis loops. We assume that the isotropic spins on A-sites quickly reach thermal equilibrium with the neighboring B-site spins, whereas the dynamics of the Ising spins occurs on a longer time scale and can be described by the Glauber dynamics[39]. To simulate hysteresis curves, we employ Glauber dynamics, a Markov chain Monte Carlo algorithm used to study non-equilibrium physics of Ising models. For each curve, the initial state is prepared with simulated annealing starting from a high-temperature ($T = 90$ K) random state. At each magnetic field value, we performed 50 measurements with $10^7$ Glauber steps in-between after the waiting time of $5 \cdot 10^8$ steps with no measurements. The total number of field points is 32. The results were averaged over 20–30 disorder realizations. Open boundary conditions in all three directions were used to speed up the magnetization reversal. The non-magnetic Zn impurities were modeled by $S_i = 0$ on randomly chosen A-sites. We simulated the lattice of $20 \times 20 \times 20$ Ising spins.

## Data availability
The data that support the key findings of this work are available in the main paper and in the Supplementary Information files or from the corresponding author upon reasonable request. Experimental data underlying Figs. 1e, 2 and 3 have been deposited in GitLab repository (https://git.rz.uni-augsburg.de/gharasom/magnetization-reversal-through-an-antiferromagnetic-state). Theoretically obtained data of Fig. 4 have been uploaded in GitHub repository (https://github.com/EBarts/Glauber_Spin_Dynamics).

## Code availability
The code used in this study is available at GitHub (https://github.com/EBarts/Glauber_Spin_Dynamics).

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

## Acknowledgements

This research was partly funded by the Deutsche Forschungsgemeinschaft (DFG, German Research Foundation)-TRR 360-492547816. The support via the project ANCD 20.80009.5007.19 (Moldova) is also acknowledged. E.B. and M.M. acknowledge Vrije FOM-programma 'Skyrmionics' and the Peregrine high performance computing cluster.

## Author contributions

L.P. and V.T. synthetized and characterized the crystals; S.G. and L.P. performed magnetization and polarization measurements and analyzed the data; D.K., K.V., and J.D. performed the THz measurements and analyzed the data; E.B. and M.M. performed the theoretical simulations; S.G., E.B., M.M., I.K., and J.D. wrote the paper; J.D. planned and coordinated the project. All authors contributed to the discussion and interpretation of the experimental and theoretical results and to the completion of the paper.

## Funding

## Competing interests

The authors declare no competing interests.
