## [Peer Review File · Nature Communications]

Magnetization reversal through an antiferromagnetic stateREVIEWER COMMENTS

Reviewer #1 (Remarks to the Author):

Ghara et al. reported re-emergence of the AFM phase that mediate the switching of H-induced ferrimagnetic order in polar magnet Zn-doped Fe₂Mo₃O₈. This looks interesting. However, there are remaining quite a few questions that should be clarified.

1, The authors should explain the equations (1) and (2) in more details. In the present stage, it looks quite intriguing. What's the difference between the current system and the traditional ferrimagnets. Especially, the fitting results following the equations give an excellent agreement with that of the THz measurements, which looks too perfect. I just wonder if it is reliable to quantitatively estimate the fraction of AFM order from the THz measurements?

2, The ferrimagnetic phase can be stabilized by a critical field smaller than the coercive field. According to the analysis given of the authors, a higher magnetic field could then drive the system back to the ground state, hard to understand.

3, In Fig. 4c, the system can be nearly completely switched to AFM. However, from the THz data, the AFM modes are obviously lower than that of the pristine AFM state. In fact, from the snapshots, the multiple domain structure is clear, and why we need the AFM domains to bridge the ferrimagnetic switching. The authors should explain this clearly. From the simulations, a rather dispersive switching from $M < 0$ to $M > 0$ is revealed, different from the observations in experiments.

4, Please keep in mind that this is a pyroelectric material. The possible influence has been simply ignored during the estimations.

Reviewer #2 (Remarks to the Author):

S. Ghara and their colleagues experimentally discovered a new magnetization reversal pathway from a ferrimagnetic state to an antiferromagnetic state. The results are interesting, but there are several concerns that need to be addressed before I can make a decision on acceptance for publication. If these concerns are resolved, I would consider accepting the paper. Please see the details below:

- From a general perspective, the energy required for switching in an antiferromagnet is higher than that in a ferromagnet (or ferrimagnet). In Fig. 2d, it can be observed that the antiferromagnetic volume increases as temperature increases. It is reasonable to assume that a higher coercivity field could be measured in a larger antiferromagnetic volume due to the larger antiferromagnetic coupling (even if it is a metastable antiferromagnet) compared to the ferromagnetic coupling. Please investigate the correlation between these parameters and clarify the physics behind them in the revised manuscript.

- Does the linewidth of X_{afm} depend on $X_{\text{afm_max}}$? If there is a metastable state in this material that becomes antiferromagnetic at the coercivity field within a certain temperature range, then it is possible that the linewidth of X_{afm} is proportional to $X_{\text{afm_max}}$. Please address this in the revised manuscript.

- The authors also conducted micromagnetic simulations in Fig. 4. It may be possible to perform a quantitative analysis of X_{afm} (similar to Fig. 2c/d) using the simulation results. Do the simulation results show a similar tendency to the experimental results? Please discuss this in the revised manuscript.

Response to the Reviewers' comments (NCOMMS-22-51275-T):

We thank the Reviewers for their valuable comments and suggestions. A point-by-point response to all their comments and a description of the subsequent changes made in the revised manuscript are given below. We have reproduced the Reviewers' comments in Black color, followed by our responses in Blue. Corresponding changes made in the revised manuscript are given in the point-by-point response below in Red color. The Red color coding is also used in the resubmitted manuscript file.

Response to Reviewer 1:

“Ghara et al. reported re-emergence of the AFM phase that mediate the switching of H-induced ferrimagnetic order in polar magnet Zn-doped $\text{Fe}_2\text{Mo}_3\text{O}_8$. This looks interesting. However, there are remaining quite a few questions that should be clarified.”

We are glad that the Reviewer found our work interesting. We discuss below his/her comments in a point-by-point response.

(1)(a) “The authors should explain the equations (1) and (2) in more details.”

Response:

To clarify the assumptions, which lead to Eqs. (1) and (2), we rewrote the entire paragraph describing the equations and elaborated on the three dominant contributions to the magnetic-field dependent polarization. In particular, we also now refer to the Reviewer’s comment that the material is pyroelectric and describe how this has been taken into account. The new paragraph in the revised manuscript reads as follows:

Consequently, we analyze the magnetic-field dependent magnetization and polarization data assuming that the entire sample volume is distributed between three fractions only, i.e. $x_{\text{afm}} + x_{\text{fim}\uparrow} + x_{\text{fim}\downarrow} = 1$. The magnetization is then solely determined by the FiM volume fractions as $M(H) = M_S(x_{\text{fim}\uparrow} - x_{\text{fim}\downarrow})$, while the polarization bears contributions of all three magnetic volume fractions,

$$P(H) = x_{\text{afm}}P_{\text{afm}}^0 + P_{\text{fim}}^0(x_{\text{fim}\uparrow} + x_{\text{fim}\downarrow}) + \alpha H(x_{\text{fim}\uparrow} - x_{\text{fim}\downarrow}). \quad (1)$$

Since the compound is pyroelectric, the electric polarization of the magnetic field-induced FiM state is different from that in the AFM ground state and may vary with temperature for both states. To include this effect, we experimentally determine the polarization of the pristine antiferromagnet in zero magnetic field P_{afm}^0 and the corresponding polarization P_{fim}^0 of the FiM phase upon lowering the field to zero after reaching the mono-domain FiM states with $M(H) = \pm M_S$ at each temperature (see Fig. 2b). Moreover, time-reversal invariance implies that the polarization is independent of the sign of \mathbf{M} or the AFM Néel vector \mathbf{L} , which results in the first two terms in Eq. (1). The third term represents the contribution due the strong linear magnetoelectric effect of the FiM state^{23,27}, where the sign of the linear magnetoelectric coefficient α is different for the two FiM volume fractions. Upon substituting $(x_{\text{fim}\uparrow} - x_{\text{fim}\downarrow}) = M(H)/M_S$, we can extract the AFM volume fraction

$$x_{\text{afm}}(H) = \frac{1}{P_{\text{afm}}^0 - P_{\text{fim}}^0} \left[P(H) - P_{\text{fim}}^0 - \alpha H \frac{M(H)}{M_S} \right], \quad (2)$$

from our experimental data of $P(H)$ and $M(H)$. The only parameter which had to be determined by fitting the $P(H)$ curves in the FiM regime is α , which was derived in the corresponding linear regimes (a possible contribution $\propto H^2$ was found to be negligible, see Supplementary note 1).

(1)(b) “In the present stage, it looks quite intriguing. What’s the difference between the current system and the traditional ferrimagnets.”

Response:

Traditional ferrimagnets do not show an AFM phase with zero net magnetization. Due to the formation of FiM honeycomb layers in the ab plane, the overall 3D magnetic order in the present material can be both AFM and FiM depending on whether the layer magnetizations

are parallel or antiparallel along the c axis (as illustrated in Fig. 1a and 1b). In addition, the origin of ferrimagnetism in the honeycomb layers is associated with the different crystallographic environments of the tetrahedrally coordinated Fe^{2+} (A) and octahedrally coordinated Fe^{2+} (B) ions (rather than two different types of magnetic ions, as observed, e.g., in the FiM spinel FeCr_2S_4 [Sci. Rep. 4, 6079 (2014); Phys. Rev. B 104, L020410 (2021)]).

Partially, the peculiarity of the magnetic layers had been discussed already in our manuscript in the Summary. We now mention the comparison to traditional 3D ferrimagnets explicitly in the Summary, where we added the following statement:

In addition, the fact that ferrimagnetism is associated with the different crystallographic environments of A- and B-site Fe^{2+} ions clearly distinguishes our case from traditional 3D ferrimagnets, such as, e.g. the spinel FeCr_2S_4 with two different types of magnetic ions^{35–37}.

(1)(c) “Especially, the fitting results following the equations give an excellent agreement with that of the THz measurements, which looks too perfect. I just wonder if it is reliable to quantitatively estimate the fraction of AFM order from the THz measurements?”

Response:

We were also surprised and very happy to find that such a simple model based on the magnetization and polarization data describes well the results obtained from the THz data. In response to the Reviewer’s concern regarding the reliability, we estimated the experimental errors for THz-derived values of x_{afm} and added errors bars to Fig. 3d of the revised manuscript. Moreover, we added the following text to the Supplementary Note 2 to clarify why the integrated intensity ratios of the AFM and FiM modes should provide a measure for x_{afm} :

By comparison of repeated measurements at 15 and 25 K, we estimated the standard deviation for the x_{afm} values derived for the AFM and the FiM mode to be about 1% and 15%, respectively. The larger experimental uncertainty for the FiM mode stems from its larger width and the increased signal-to-noise ratio of our experimental setup in the corresponding frequency range.

Note that this evaluation procedure is based on the fact that in linear response the absorption coefficient is assumed to be proportional to the density $n_V = N/V$ of N entities interacting with the radiation in the crystal of volume V . Hence, normalizing the absorption intensity in the metastable AFM state at the coercive field to the purely AFM state at $H = 0$ yields the corresponding ratio of the numbers of interacting centers $N_{\text{afm}}(H = H_C)/N_{\text{afm}}^{\text{total}}(H = 0)$. Whether this particular AFM absorption can be ascribed to A- or B-site iron ions or to a collective excitations of all magnetic ions is not clear at present, but one can safely assume that the interacting ions are homogeneously distributed in the corresponding volume fractions and hence we identify $x_{\text{afm}} = N_{\text{afm}}(H = H_C)/N_{\text{afm}}^{\text{total}}$. A similar argument holds for the FiM excitation. The remarkably good agreement with the volume fraction x_{afm} derived from the dc-values as shown in Fig. 3d justifies our assumptions.

(2) “The ferrimagnetic phase can be stabilized by a critical field smaller than the coercive field. According to the analysis given of the authors, a higher magnetic field could then drive the system back to the ground state, hard to understand.”

Response:

Our picture concerning the stability of different magnetic phases is as follows. The AFM-to-FiM transition (i.e. spin-flip transition) occurs via flipping the layer magnetization in every second layer, while the switching between the FiM-up and FiM-down state implies flipping the magnetization of all layers. As temperature decreases, the spin-flip or critical field increases, because of the decrease of the magnetic moment in the FiM state. This reduction of the

magnetization is observed experimentally in $\text{Fe}_2\text{Mo}_3\text{O}_8$ [Phys. Rev. X 5, 031034 (2015)] and also in $\text{Fe}_{1.86}\text{Zn}_{0.14}\text{Mo}_3\text{O}_8$ and explained theoretically in our manuscript by the difference in amplitudes of spin fluctuations on Ising-like B-sites and relatively isotropic A-sites (please see also our reply to the comment 1 of Reviewer 2). Moreover, the slow kinetics at low temperatures prevents the metastable FiM state to transform into the AFM state, which is stable at low magnetic fields. The appearance of the AFM is thus "delayed" till a high (coercive) field, at which the magnetization changes sign. As a result, the AFM state only appears in a narrow field interval near the coercive field as a meta-stable state (not as the ground state).

(3)(a) "In Fig. 4c, the system can be nearly completely switched to AFM. However, from the THz data, the AFM modes are obviously lower than that of the pristine AFM state. In fact, from the snapshots, the multiple domain structure is clear, and why we need the AFM domains to bridge the ferrimagnetic switching. The authors should explain this clearly."

Response:

We thank the Reviewer for this remark. The snapshots of the simulation shown in Fig. 4c indeed show an almost complete disappearance of the FiM state in favour of the AFM state. However, this is just a single simulation. The magnetic field dependence of the polarization plotted in Figs. 4a and 4b are obtained by averaging over 30 simulations of the switching process (as described in the Methods section). One can see that the polarization at the magnetization reversal is smaller than that of the virgin state and it decreases with decreasing temperature. This is consistent with the THz data (Fig. 3b and 3c), where we observe a lower intensity of the AFM mode at the magnetization reversal than that of the virgin state due to the fact that only around 60% of the sample volume turns into the AFM state at 25 K. This is now explained more clearly in the paragraph describing Fig. 4c by the following statement:

The nearly uniform color of each plane reflects strong spin correlations in the *ab* layers (see Fig. 4c). These spin correlations suppress the growth of droplets with the opposite magnetization induced by the magnetic field reversal. The correlation length along the *c*-axis in the AFM phase is small due to weak interlayer coupling. The slow kinetics of the magnetization switching and the enhanced stability of the AFM phase at low temperatures delay the emergence of the AFM state up to the coercive field. The simulation snapshot in Fig. 4c shows almost complete disappearance of the FiM state in favor of the AFM state, indicating the importance of the small interlayer coupling. Averaged over many simulations, the maximal fraction of the AFM phase decreases with decreasing temperature (see Figs. 4a,b) in agreement with our polarization and THz data.

(3)(b) "From the simulations, a rather dispersive switching from $M < 0$ to $M > 0$ is revealed, different from the observations in experiments."

Response:

As pointed out by the Reviewer, the experimental data may imply a somewhat larger correlation length along the *c*-axis as compared to the theoretical picture, where a nearly stochastic switching of individual layers happens. The width of the magnetization reversal region in our simulations depends on the number of Glauber spin-flip steps for a given magnetic field value and the number of Fe sites. To simplify the modelling, we used a fixed number of steps in all our simulations (as explained in Methods). We could have made this number temperature-dependent and adjust it to fit the shape of the magnetic field dependent magnetization and polarization curves. We think, however, that such procedure would add little to the understanding of the switching process (see also our reply to the comment 3 of Reviewer 2).

In response to the comments of Reviewer 2, we added a new Supplementary Fig. 3 and added the full-width-half-maximum value of the $x_{afm}(H)$ curves to Fig. 2d, which is described as follows in the Supplementary note 1:

This is clearly shown in Supplementary Fig. 3, where we plot the magnetic field-dependent x_{afm} for three selected temperatures. This figure directly shows that, upon lowering temperature, the maximum value x_{afm}^{max} decreases, while the width of the symmetric distribution increases. In our numerical simulations, x_{afm}^{max} also decreases, whereas the width of the magnetization reversal region is approximately constant, as it is determined by the number of Glauber spin-flip steps for a given magnetic field value, which for simplicity we keep temperature-independent.

(4) “Please keep in mind that this is a pyroelectric material. The possible influence has been simply ignored during the estimations.”

Response:

Since the present material is pyroelectric, the electric polarizations of the AFM and FiM phases are different, which is taken into account in analysing our experimental data using Eq.(1) (see also our reply to the Reviewer’s comment 1 and the corresponding changes in the text.). Since our numerical calculations of the temperature and field dependence of the electric polarization (see Supplementary figures 5b, 6b and 7b) are in good agreement with experiment, we neglected this additional variation of the polarization in the two states with temperature.

Response to Reviewer 2:

“S. Ghara and their colleagues experimentally discovered a new magnetization reversal pathway from a ferrimagnetic state to an antiferromagnetic state. The results are interesting, but there are several concerns that need to be addressed before I can make a decision on acceptance for publication. If these concerns are resolved, I would consider accepting the paper. Please see the details below:”

Response:

We thank the Reviewer for finding our work interesting. We address all the Reviewer’s comments in our reply below.

(1) “From a general perspective, the energy required for switching in an antiferromagnet is higher than that in a ferromagnet (or ferrimagnet). In Fig. 2d, it can be observed that the antiferromagnetic volume increases as temperature increases. It is reasonable to assume that a higher coercivity field could be measured in a larger antiferromagnetic volume due to the larger antiferromagnetic coupling (even if it is a metastable antiferromagnet) compared to the ferromagnetic coupling. Please investigate the correlation between these parameters and clarify the physics behind them in the revised manuscript.”

Response:

Although the maximal fraction of the AFM phase is close to 1 at high temperatures and decreases with decreasing temperature, the critical field of the transition from the AFM to FiM state strongly increases as temperature is lowered. This effect was experimentally observed both in pure $\text{Fe}_2\text{Mo}_3\text{O}_8$ and Zn-doped $\text{Fe}_2\text{Mo}_3\text{O}_8$ [Phys. Rev. X 5, 031034 (2015); Phys. Rev. B 102, 174407 (2020)]. We explain this effect by the difference in amplitudes of spin fluctuations on tetrahedrally coordinated A and octahedrally coordinated B sites. At high temperatures, stronger fluctuations of isotropic Fe^{2+} spins in A-site make their relatively smaller magnetic moment even smaller, thus increasing the net magnetic moment of FiM ab layers and lowering the spin-flip field. As temperature goes down, spin fluctuations become suppressed, the saturated magnetic moment in the FiM state decreases and a higher applied field is necessary to transform the AFM state into the FiM state. This enhanced resilience of the AFM state to magnetic fields is important for the intervention of this state at the coercive field. These aspects have already discussed in the original manuscript in page 5-6. In addition, as mentioned in our reply to the comment 2 of Reviewer 1, the kinetics of phase transformations also plays an important role: at low temperatures, it slows down the transition between the metastable FiM and stable AFM states at low magnetic fields. This is now mentioned in the discussion of the theoretical results (see changes in response to comment 3 of Reviewer 1).

(2) “Does the linewidth of x_{afm} depend on $x_{\text{afm}}^{\text{max}}$? If there is a metastable state in this material that becomes antiferromagnetic at the coercivity field within a certain temperature range, then it is possible that the linewidth of x_{afm} is proportional to $x_{\text{afm}}^{\text{max}}$. Please address this in the revised manuscript.”

Response:

We thank the Reviewer for pointing out the importance of the width of $x_{\text{afm}}(H)$. We evaluated the full-width-half-maximum values of the curves and added them to the data of $x_{\text{afm}}^{\text{max}}$ in Fig. 2d in the revised manuscript. The width increases linearly with decreasing temperature. Moreover, we show a direct comparison of $x_{\text{afm}}(H)$ curves at three different temperatures as a new separate figure in the Supplementary Note 1, which is described as follows in the Supplementary Materials:

This is clearly shown in Supplementary Fig. 3, where we plot the magnetic field-dependent x_{afm} for three selected temperatures. This figure directly shows that, upon lowering temperature, the maximum value $x_{\text{afm}}^{\text{max}}$ decreases, while the width of the symmetric distribution

increases. In our numerical simulations, x_{afm}^{max} also decreases, whereas the width of the switching interval is approximately constant, as it is determined by the number of Glauber spin-flip steps for a given magnetic field value, which for simplicity we keep temperature-independent.

As a consequence of the linear behavior of both quantities on temperature the direct correlation of the two parameters is also linear as shown in the figure below. We believe that the linear behavior of both quantities with temperature may be a signature of the different kinetics of the metastable states, which we explicitly state now in the revised manuscript as follows:

Figure 1: Plot of x_{afm}^{max} vs. the full-width-half-maximum values ΔH_{afm} for various temperatures.

The corresponding maximum values x_{afm}^{max} and the full-width-half-maximum width ΔH_{afm} of the curves are shown in Fig. 2d for all investigated temperatures (see supplementary Fig. 3 for details). While x_{afm}^{max} decreases linearly with decreasing temperature and extrapolates to zero at around 10 K, the linear increase of ΔH_{afm} may reflect different kinetics of the metastable states in the vicinity of the magnetization reversal.

(3) “The authors also conducted micromagnetic simulations in Fig. 4. It may be possible to perform a quantitative analysis of x_{afm} (similar to Fig. 2c/d) using the simulation results. Do the simulation results show a similar tendency to the experimental results? Please discuss this in the revised manuscript.”

Response:

Our simulations demonstrate the appearance of the AFM state at the magnetization reversal, as well as the growth of the critical field with decreasing temperature. Consistent with the experimental results, the maximal volume fraction of the AFM state in our simulations decreases with decreasing temperature (although not as fast as in experiment), as indicated by the reduced polarization at the magnetization reversal with lowering temperature (Fig. 4a and 4b).

The simulations, however, cannot describe all aspects of the kinetics of phase transformations. For example, the width of the magnetization reversal region is temperature independent in the simulation. The reason is the relatively small system size that we can handle numerically. To reproduce the phase diagram of Zn-doped $Fe_2Mo_3O_8$, we use model parameters appropriate for the bulk material, and in particular, the small value of the interlayer spin-spin interaction J_{\perp} . For the $20 \times 20 \times 20$ lattice of B-site spins studied numerically, the spin correlations along

the c direction are then short-ranged and the transition from the FiM to AFM state occurs layer-by-layer, as shown in Fig 4d.

In macroscopic bulk samples the kinetics of the magnetization switching and transformations between the FiM and AFM phases is three-dimensional (despite the weak interlayer interactions) and involves thermally activated motion of domain walls pinned by impurities and lattice imperfections. We think that the domain wall dynamics is, in particular, responsible for the decrease of x_{afm} with decreasing temperature, since the large energy difference between FiM-up and FiM-down states at high switching fields exerts a strong force on the domain walls and favors the direct magnetization switching, not mediated by the AFM state. To describe such processes, one would need to do simulations on much bigger spin lattices, which is not feasible, or to increase the interlayer coupling, which would make the system effectively three-dimensional, but at the same time it will lead to unphysically high spin-flip fields. For the same reason, we decided not to use the number of Glauber steps as a temperature-dependent parameter (see our reply to comment 3 of Reviewer 1), since this would not improve the modelling of the switching kinetics.

This is now discussed in the paragraph describing the simulation results of Fig. 4. as follows:

The nearly uniform color of each plane reflects strong spin correlations in the ab layers (see Fig. 4c). These spin correlations suppress the growth of droplets with the opposite magnetization induced by the magnetic field reversal. The correlation length along the c -axis in the AFM phase is small due to weak interlayer coupling. The slow kinetics of the magnetization switching and the enhanced stability of the AFM phase at low temperatures delay the emergence of the AFM state up to the coercive field. The simulation snapshot in Fig. 4c shows almost complete disappearance of the FiM state in favor of the AFM state, indicating the importance of the small interlayer coupling. Averaged over many simulations, the maximal fraction of the AFM phase decreases with decreasing temperature (see Figs. 4a,b) in agreement with our polarization and THz data.

REVIEWERS' COMMENTS

Reviewer #1 (Remarks to the Author):

I think the authors have addressed the comments and suggestions from the previous report properly, and I recommend the paper for publication.

Reviewer #2 (Remarks to the Author):

I sincerely appreciate the author's dedicated efforts. Through the initial revision process and subsequent rebuttal, all of my concerns have been effectively addressed. Therefore, I am pleased to highly recommend the publication of this work in Nature Communications.

Response to the Reviewers' comments (NCOMMS-22-51275-A):

We thank the Reviewers for their valuable comments and suggestions throughout the revision process, and we are happy to know that both the Reviewers find our revised version suitable for publication in Nature Communications.

We mention below a point-by-point response to their comments. We have reproduced the Reviewers' comments in Black color, followed by our responses in Blue.

Response to Reviewer 1:

"I think the authors have addressed the comments and suggestions from the previous report properly, and I recommend the paper for publication."

We thank the Reviewer for recommending our manuscript for publication in Nature Communications.

Response to Reviewer 2:

"I sincerely appreciate the author's dedicated efforts. Through the initial revision process and subsequent rebuttal, all of my concerns have been effectively addressed. Therefore, I am pleased to highly recommend the publication of this work in Nature Communications."

We are glad to know that our efforts in revising the manuscript successfully met the Reviewer's expectations. We thank the Reviewer for his/her highly positive assessment on our work.